# A Review on Gel Polymer Electrolytes for Dye-Sensitized Solar Cells

**DOI:** 10.3390/mi13050680

**Published:** 2022-04-27

**Authors:** Prasad Raut, Vinay Kishnani, Kunal Mondal, Ankur Gupta, Sadhan C. Jana

**Affiliations:** 1School of Polymer Science and Polymer Engineering, The University of Akron, Akron, OH 44325, USA; prsdraut@gmail.com (P.R.); janas@uakron.edu (S.C.J.); 2Department of Mechanical Engineering, Indian Institute of Technology Jodhpur, Jodhpur 342037, Rajasthan, India; kishnani.1@iitj.ac.in; 3Idaho National Laboratory, Idaho Falls, ID 83415, USA

**Keywords:** DSSC, liquid electrolytes, solar cells

## Abstract

Significant growth has been observed in the research domain of dye-sensitized solar cells (DSSCs) due to the simplicity in its manufacturing, low cost, and high-energy conversion efficiency. The electrolytes in DSSCs play an important role in determining the photovoltaic performance of the DSSCs, e.g., volatile liquid electrolytes suffer from poor thermal stability. Although low volatility liquid electrolytes and solid polymer electrolytes circumvent the stability issues, gel polymer electrolytes with high ionic conductivity and enduring stability are stimulating substitutes for liquid electrolytes in DSSC. In this review paper, the advantages of gel polymer electrolytes (GPEs) are discussed along with other types of electrolytes, e.g., solid polymer electrolytes and p-type semiconductor-based electrolytes. The benefits of incorporating ionic liquids into GPEs are highlighted in conjunction with the factors that affect the ionic conductivity of GPEs. The strategies on the improvement of the properties of DSSCs based on GPE are also presented.

## 1. Introduction

Recently published market surveys suggest that the world’s energy demand will increase by almost 50% from 2018 to 2050 [1]. Fossil fuels that supply an approximately major percentage of the energy disbursed over the world are facing a rapid exhaustion of these resources. As per the statistical review of world energy, the world resource reserves of fossil fuels in 2016 were anticipated to last around 50 years and ~115 years for oil/natural gas and for coal, respectively [2]. There are burgeoning requirements for environmentally viable energy technologies compounded by the growing demand for energy, exhaustion of fossil resources, global warming and concomitant climate fluctuations. Wind turbines, wave and tidal power, hydropower, biomass-derived liquid fuel solar cells, solar thermal, and biomass-fired electricity generation are among the most prevalent renewable energy technologies; among these, photovoltaic technology is the most promising. Fortunately, the sun’s energy supply to the globe is enormous: 1.25 × 10^24^ Cal per year, or few thousand times more than the world’s current consumption. A simple calculation supports that solar cells with an efficiency of 10% covering 0.1% of the earth’s surface can meet our present energy needs [3].

The basic working principle of solar cells lies in the fact that it utilizes the energy generated by the sun by converting solar radiation directly into electricity. The first practical conversion of the radiation into electric energy was validated by Bell Telephone Laboratories in 1954, and produced 6% efficiency by using p–n junction-type solar cells [4]. The photovoltaic cells made from semiconductor-grade silicon quickly became the power source of choice for use on satellites, and were catalyzed by the initiation of the space program. The common solar power conversion efficacies fall in the range of around 15–20% [5]. However, the relatively high cost of silicon cells, as well as the usage of harmful chemicals in their manufacture, are deterrents to their widespread adoption. These factors prompted researchers to look for inexpensive, ecologically responsive solar cell alternatives.

Solid-state junction devices, which arose from semiconductor industry experience in materials research, currently dominate the sector. On the other hand, the supremacy of inorganic solid-state junction devices is being defied by a third generation of cells based on nanocrystalline and conducting polymer films, for example. These novel materials promise low-cost production and have appealing qualities that make market entry easier. It is now possible to entirely abandon the traditional solid-state junction device and replace the contacting phase with an electrolyte derived from liquids, gels, or solids, resulting in the formation of a photo-electrochemical cell. However, the value of reporting power conversion efficiency as a function of absorber material bandgap for the key new photovoltaic technologies: perovskite, organic, and dye-sensitized solar cells have been briefly discussed [6,7].

The recent and impressive advances in the production and characterization of nanocrystalline materials have opened up a plethora of new possibilities. Devices based on interpenetrating networks of mesoscopic semiconductors have shown astonishingly high conversion efficiencies that rival those of conventional devices, contrary to expectations. Dye-sensitized solar cells are the prototype of this class of devices, which achieve optical absorption and charge separation by combining a sensitizer as a light-absorbing material with a wide bandgap semiconductor with a nanocrystalline morphology [8].

Gratzel et al. originally described the dye-sensitized solar cells (DSSCs) made using nanocrystalline TiO_2_ based on the principle of a fast regenerative photo-electrochemical process in 1991 [8]. This novel type of solar cell had an overall efficiency of 7.1–7.9% (under simulated solar light), which is comparable to amorphous silicon solar cells [3]. The separation of the functional-light-absorbing “dye” from the charge carrier transport in the former is the fundamental distinction between this type of solar cell and ordinary cells. This feature allows for the DSSCs to work with low- to medium-purity, ecologically friendly materials while maintaining commercially viable energy conversion efficiency.

A translucent electrode coated with a dye-sensitized mesoporous layer of nanocrystalline particles of TiO_2_, an electrolyte containing an appropriate redox couple and a Pt-coated counter-electrode are often used in Gratzel cells. Because the nanocrystalline DSSCs are made up of many materials, the qualities of each component have a direct impact on the kinetics and reactions, and hence the solar cell’s performance. As a result, the device’s performance is influenced by the porous semiconductor film’s structure, morphology, optical and electrical properties, as well as the dye’s chemical, electrochemical, photophysical, and photochemical properties, the electrochemical and optical properties of the redox couple and solvent in the electrolyte, and the electrochemical properties of the counter-electrode [9].

DSSCs based on Ru-bipyridyl complexes and liquid electrolytes can have up to 11 percent efficiency. However, there are certain apprehensions regarding the existence of the liquid component, which necessitates proper sealing to avoid leakage. The cells’ shape and steadiness are limited as a result of the sealing requirement. Many research groups have concentrated on replacing liquid electrolytes with solid or gel-type electrolytes in dye-sensitized solar cells to achieve reduced costs and easier construction [10]. Inorganic or organic hole conductors, gel electrolytes, gel electrolytes generated with ionic liquids or by the solidification of liquids, and polymer electrolytes are the principal alternative materials [10]. An overview of recent developments in dye-sensitized solar cells assembled with polymer and gel electrolytes is presented in this review article, with a focus on the modifications made to improve the ionic conductivity and the mechanical stability of such materials, as well as how such modifications affect the performance of polymer-based DSSCs.

## 2. Dye-Sensitized Solar Cell Architecture and Basic Operation

A schematic of the interior shown in Figure 1 can be used to understand the operating principle of a DSSC.

The following is a typical DSSC configuration: the mesoporous oxide layer, which is made up of a network of TiO_2_ nanoparticles sintered together to form the electronic conduction channel, is at the heart of the device. The film thickness is normally around 10 µm and is made up of nanoparticles with diameters ranging from 10 to 30 nm. The film’s porosity ranges from 50 to 60%. Atop a glass or plastic substrate, the mesoporous layer is formed on a transparent conducting oxide (TCO) [10]. As shown in Figure 1, a common substrate is glass-covered with fluorine-doped tin oxide (FTO). On the surface of the nanocrystalline film, the charge-transfer dye is placed as a monolayer. When a dye is photoexcited, an electron is injected into the oxide’s conduction band, leaving the dye in its oxidised state. Electron transfer from the electrolyte, which is commonly an organic solvent containing the iodide/triiodide redox system, returns the dye to its ground state.

The iodide intercepts the oxidised dye’s recapture of the conduction band electron, causing the sensitizer to regenerate. The $I_{3}^{-}$ ions produced by the oxidation of $I^{-}$ diffuse a short distance (<50 µm) through the electrolyte to the cathode, which is coated with a thin coating of platinum catalyst, where electron transfer completes the regenerative cycle by reducing $I_{3}^{-}$ to $I^{-}$. The major steps for converting photons to current are as follows:The incident photon is absorbed by the photosensitizers in the Ru complex that are adsorbed on the TiO_2_ surface. From the ground state (*S*) to the aroused state (*S**), photosensitizers are excited.
(1)$$S+h\nu \;\to S^{*}$$

2.The excited electrons are injected into the TiO_2_ electrode’s conduction band. The photosensitizer (*S*^+^) gets oxidised as a result of this interaction.


(2)
$$S*\to S^{+}+e^{-}\left({\mathrm{TiO}}_{2}\right)$$


3.The injected electrons in TiO_2′_s conduction band diffuse between nanoparticles, eventually reaching the back contact (TCO). Through the circuit, the electrons eventually reach the counter-electrode.4.The oxidized photosensitizer (*S^+^*) receives electrons from the I^−^ ion redox mediator, resulting in the ground state (*S*) being regenerated and the I^−^ being oxidised to the oxidised state, I_3_^−^.


(3)
$$S^{+}+e^{-}\to S$$


5.The oxidized redox mediator, I_3_^−^, diffuses toward the counter-electrode and then it is reduced to I^−^ ions.


(4)
$$I_{3}^{-}+2e^{-}\to 3I^{-}$$


O’Regan and Durrant [11] gave the following details on typical materials and relative concentrations of different species in the mesoporous system under regular working conditions:Each TiO_2_ particle has roughly ~10 electrons in the operating conditions;In TiO_2_, more than 90% of electrons are restrained, with only <10% in the conduction band;On an 18 nm TiO_2_ particle, there are approximately 10,000 H^+^ adsorption sites;On the surface of a TiO_2_ particle (18 nm), there are approximately 600 dye molecules;Every second, a photon is absorbed by each dye molecule;Injection of electrons into TiO_2_ particles occurs at a rate of approximately 600 s^−1^;Under normal operating conditions, around 1 dye in every 150 TiO_2_ particles becomes oxidized;In the electrolyte, the total volume percentage of the solutes is approximately 10–20%;There will be approximately 1000 $I^{-}$ and 200 $I_{3}^{-}$ ions in the pore volume around the TiO_2_ particle;Iodine, I_2_, has a concentration of <1 µM, or about one free iodine per 10,000 TiO_2_ particles.

The required turnover number for a DSSC to be durable for more than 15 years in outdoor installations is 10^8^, which can be met by ruthenium complexes [12]. The difference in the electrochemical potential of the electron at the two contacts corresponds to the voltage created under light. The difference between the Fermi level of the mesoporous TiO_2_ layer and the redox potential of the electrolyte is the difference in DSSC. Overall, no permanent chemical reaction is required to generate electric power. Figure 2 depicts the basic electron transfer mechanisms in a DSSC, as well as the potentials for a cutting-edge device based on N_3_ dye adsorbed on TiO_2_ and $I^{-}/I_{3}^{-}$ as the redox couple in the electrolyte.

The loss reactions 1, 5, and 6 are shown in Figure 2 in addition to the desired pathway of the electron transfer processes (processes 2, 3, 4, and 7 in Figure 2) described above. The excited-state lifetime reflects a direct recombination of the excited dye in reaction 1. The recombination of injected electrons in TiO_2_ with oxidized dyes or acceptors in the electrolyte is numbered 5 and 6.

In theory, electron transport to $I_{3}^{-}$ can happen at the interface between the nanocrystalline oxide and the electrolyte, or at exposed portions of the anode contact (typically a fluorine-doped tin oxide layer on glass). In practice, the second channel can be blocked by spray pyrolysis, depositing a dense blocking layer of oxide on the anode [13,14]. For DSSCs that use one-electron redox systems or cells that use solid organic hole-conducting mediums, blocking layers are required [15,16].

Hundreds of alternatives to the components utilized in traditional DSSCs have been studied, as previously stated. When it comes to sensitizers, Ru-complexes have been the most effective since the beginning. Other organometallic compounds, such as phthalocyanines and porphyrins, as well as osmium and iron complexes, have been developed. Metal-free organic dyes are trying to catch up, with indoline dyes demonstrating efficiencies of around 10% [17,18]. Moreover, numerous groups have recently created chemically resistant organic dyes with promising stability results [19,20,21,22]; the references also include recent overviews of photoanode materials [23,24,25], TiO_2_, ZnO, SnO_2_, and Nb_2_O_5_ are the most popular oxides. Nanoparticles, nanofibers and tubes, and core–shell structures all have been used to create new morphologies. A platinized conducting glass is the most typical counter-electrode. In addition, conductive polymers and carbon compounds were also produced [26,27,28]. Further, it was observed that the introduction of π-extended dibenzo-BODIPY into organic sensitizers improves the power conversion efficiency in DSSC [29]. BODIPY dyes have exceptional characteristics, particularly near IR sensitizers. Modifications to improve these dyes’ performance in other areas of the solar spectrum will make them very promising as similar to the solar cell sensitizer dyes [30].

## 3. Electrolytes

Gratzel and O’Regan [8] published a report on a mixed solvent electrolyte system in 1991, consisting of 80:20 ethylene carbonate and acetonitrile by volume. A combination of 0.5 M tetrapropylammonium iodide and 0.04 M iodine was used as the redox component. The study reported a conversion efficiency of 7.9%. The electrolyte composition was adjusted by adding low concentrations of lithium or potassium iodide without affecting the conversion efficiency. After decades of intensive study employing a variety of alternate solvents, redox couples, and various additions, the same categories of nanoparticles, dyes, and electrolytes are now used. Although the corrosive and photochemical properties of iodine are less than ideal, and new research on alternative redox couples is underway, the electrolyte based on the $I^{-}/I_{3}^{-}$ redox couple has been a preferred choice as the hole-conducting medium. Alternative redox systems, such as cobalt-based systems, SCN^−^/(SCN)^3−^ and SeCN^−^/(SeCN)^3−^, have shown promising results in recent investigations. It is reported that a tris(2,2′-bipyridine)cobalt(ΙΙ)/(ΙΙΙ)-based gel polymer electrolyte shows an exceptional energy conversion efficiency of 8.7% and 10% under 1 sun and 0.1 sun, respectively, for a stable DSSC [31].

Furthermore, DSSCs of the Co(ΙΙ)/Co(ΙΙΙ) complex were fabricated through the in-situ process and it was observed that the efficiency of power conversion had been exceeded up to 6.5% after 1800 h and up to 8.5% at low intensity [32]. As an alternative to (iodine-based) redox systems, poly(oxyethylene)-imide-imidazolium selenocyanate (POEI-IS) has been used for a versatile gel electrolyte DSSC. It contains various functions, viz. gelling agent, redox mediator of SeCN^−^, and formed a chelate with potassium cations [33]. Another alternate for the $I^{-}/I_{3}^{-}$ redox in DSSC that offers an attractive alternate is [Co(bpy-pz)_2_]^3+^/^2+^(PF_6_)_3/2_. It has a power conversion efficiency of more than 10% [34].

In comparison to other ionic liquids, 1-ethyl-3-methylimidazolium selenocynate (EMISeCN) is said to have a low viscosity. As a result of the weakening of the van der Waals forces associated with the highly polarizable iodide component, it retains better conductivity due to its low cohesive energy [35,36]. Co-grafting on hexa-decylmalonic acid (HDMA) was used to improve the photovoltaic performance in another study. When the two amphiphiles are co-grafted, a varied monolayer is formed, which should be more tightly packed than when the sensitizer is adsorbed alone, producing a more efficient insulating barrier for back-electron transfer [37]. Furthermore, the trends that enable iodide-free redox couples as being the most successful, as well as their viability for use in DSSCs, utilizing fresh and novel photosensitizer and counter-electrode materials briefly discussed [38].

## 4. Liquid Electrolyte

The $I^{-}/I_{3}^{-}$ redox system is dissolved in a suitable solvent to form a liquid electrolyte. Diffusive mass movement of charge carriers in the electrolyte is a critical parameter for stable cell operation and optimal solar power generation. The transport mechanism is influenced by the ions’ diffusion coefficient, the solvent’s viscosity, and the porous film electrode’s structure. The electrolyte’s solvent allows charge carriers to diffuse quickly and prevents dye desorption from the oxide surface during the redox reaction. Acetonitrile, ethyl carbonate, and some other carbonates, viz. dimethyl carbonates, diethyl carbonates, ethylene carbonates, propylene carbonates to name a few, are the commonly used solvents although their usage is fraught with issues of poor sealing, thermal degradation, and safety as these small molecule solvents easily escape into air due to their high volatility. A complete seal is necessary to prevent the loss of liquid solvents from the electrolyte system due to leakage and/or evaporation. The liquid junction DSSCs in this case require a flawless seal with a binder that is chemically resistant to the electrolyte [39]. These disadvantages obstruct cell manufacturing; in particular, the use of liquid electrolytes obstructs the large-scale application of DSSCs while also limiting the shape and stability of the cells if a high-speed, roll-to-roll continuous manufacturing method is used for industrial DSSC manufacturing [10,40].

Some research groups concentrated on replacing liquid electrolytes with inorganic or organic whole conductors and polymer electrolytes, which reduces the cost of dye-sensitized solar cells and makes construction easier. Solid polymer electrolytes and gel polymer electrolytes are two types of polymer electrolytes (GPEs).

## 5. P-Type Semiconductors

The p-type DSSC is made up of a photoactive working electrode (cathode), a passive counter-electrode (anode), and a redox electrolyte in a sandwich shape. In p-type DSSCs, the dye absorbs visible light and then transfers electrons from the semiconductor’s valence band to the dye. The dye is subsequently regenerated in the electrolyte by electron transfer from the reduced dye to the oxidised species. It is possible that the decreased dye will reunite with the hole in the semiconductor if it can’t react with the electrolyte within the charge separated lifespan. The holes in the semiconductor migrate to the working electrode’s back collector, while the electrolyte’s reduced species diffuse to the electrode. In the external circuit, this charge collection causes a cathodic photocurrent. Solid-state electrolytes are primarily thought of as materials that transmit holes (HTM). If a material with p-type semiconducting activity absorbs holes from the dye cation, it can possibly replace the liquid electrolyte in DSSCs. However, in liquid electrolyte-based DSSCs, the transport mode shifts from ionic to electronic transport in HTM-based solid-state DSSCs. Traditional HTMs are inorganic p-type materials with increased hole mobility, such as CuI and CuSCN. When they are directly utilized in DSSCs, however, their crystallization rate is quick, and control of crystal size and growth rate is challenging to maintain, resulting in incomplete filling of TiO_2_ pores, as presented in Figure 3a. As a result, the efficiency may be less than 1% [41]. As shown in Figure 3b, molten salts such as 1-methyl-3-ethylimidazolium thiocyanate and triethylaminehydrothiocyanate may effectively limit CuI crystal development and facilitate filling of the pore of dyed TiO_2_ anode [42] resulting in a 3.8 percent boost in light-to-electrical efficiency.

Furthermore, naphthalene imides are reported as a novel p-type sensitizer for NiO-based DSSC. These two DSSCs, namely S64 and S85 with prolonged π-conjugations and lengthy alkyl chains have good solubility in organic solvents. In a NiO-based p-type dye-sensitized solar cells, these dyes have a high efficiency. Their exterior quantum efficiency measurements also revealed a reasonable efficiency in the visible range [43]. In another work, π-extended dibenzo-BODIPY sensitizer with triphenylamine and nitrothiophene synthesized, which showed an intense band of absorption at 730 nm. It was observed that the performance of this NiO-based p-type DSSC was low due to the very fast recombination of NiO and dye at the surface of the electrode [44].

The organic molecular solids and polymers provide attractive diversity comparable to inorganic HTMs and in conjunction are amenable to chemical modifications to fit different needs. Examples include polypyrrole, polythiophene, and polyaniline, to name a few, as shown in Figure 4. These materials exhibit a good balance of electrical, electronic, and optical properties of metals and semiconductors, and mechanical flexibility of conventional polymers. Prior work demonstrated their applications in solid-state DSSCs [45]. Despite considerable progress, the low conversion efficiency of solid-state DSSCs remains a challenge. Solid conductive materials’ penetration into semiconductor porous films is still low, and organic HTM conductivity is limited by diffusion.

## 6. Solid Polymer Electrolytes

The study of polymer electrolytes commenced in the 1970s after Wright and colleagues reported their studies on ionic conductivity in polymer–salt compositions [46]. Secondary batteries took advantage of these systems. After Wright’s work [46], Polyethers, such as poly(ethylene oxide) (PEO), in combination with a variety of inorganic salts, such as LiI, NaI, LiClO_4_, LiCF_3_SO_3_, LiSCN, NaClO_4_, or LiPF_6_, have become the standard systems for further investigation. The repeating unit (–CH_2_–CH_2_–O–) in PEO provides a promising configuration for active interactions between the free electron pair in oxygen and the alkali metal cations. Because the PEO chains are organized in a helical shape that are hollow, the optimum distances for oxygen–cation interactions are created. At temperatures between 40 and 100 °C, PEO–salt complexes typically have conductivities in the range of 10^−8^ to 10^−4^ S/cm [8], restricting their use at room temperature. The solid-state nature of polymer electrolytes is advantageous; however, the ionic conduction in the amorphous phase of most polymer electrolytes is insufficient for photo-electrochemical cell applications. A specific degree of disorder must be induced in the structure to minimize the degree of crystallinity of the polymer at ambient temperature and, therefore, boost ionic mobility. This can be accomplished by combining various polymers, copolymers, or cross-linked networks to lower the glass transition temperature or diminish the crystallinity of the polymer. A third component, which can operate as a plasticizer, can also be introduced into the system [47]. Ionic mobility in polymer electrolytes is intimately linked to local structural relaxations that occur in the amorphous phase. The ionic conductivity may easily be modified to further increase gadget performance. In this context, adding inorganic nanofillers, ionic liquids, ethylene oxide oligomers, plasticizers, and other additives to create polymer (or gel) electrolytes with increased ionic conductivity qualities has become a typical technique [10].

## 7. Gel Polymer Electrolytes (GPEs)

GPEs are made by trapping liquid electrolytes that contain organic solvents and inorganic salts such as ethylene carbonate (EC), propylene carbonate (PC), or sodium iodide (NaI), acrylonitrile (ACN), lithium iodide (LiI), and potassium iodide (KI). The value of short-circuit density (Jsc) decreases in systems with GPEs due to gelation, but the open-circuit voltage (Voc) rises due to the suppression of a dark current by polymer chains covering the TiO_2_ electrode’s surface [48]. These tendencies combine to give DSSCs with GPEs nearly the same efficiency (η) as those with liquid electrolytes. Quasi-solid-state DSSCs are cells that were built utilizing GPEs.

## 8. The Advantages of GPE

The GPEs are made by encasing a liquid electrolyte in polymer cages. Some of the benefits of GPEs are their low vapour pressure, superior wetting and filling properties between the nanostructured electrode and counter-electrode, higher ionic conductivity than typical polymer electrolytes, and outstanding thermal stability. As evidenced by the wide range of applications, these characteristics lead to the remarkable long-term stability of the DSSCs [49,50,51].

Because of their liquid state over a broad temperature range, non-flammability, and low vapour pressure at room temperature, wide electrochemical windows, high ionic conductivity, as well as excellent thermal and chemical stability, a large number of published reports on GPEs in the last decade have focused on ionic liquids (ILs) [52,53]. Kubo et al. [49] developed a DSSC based on room-temperature molten salt. These authors investigated the physical–electrochemical properties of 1-hexyl-3-methylimidazolium iodide (HMImI) and its mixtures with organic solvents, such as acetonitrile, and with other lower viscosity ILs, such as 1-ethyl-3-methylimidazoliumtriflate (EMImTf). Furthermore, it was proposed based on the data on diffusion coefficients of $I_{3}^{-}$ in pure HMImI that an electron exchange via a Grotthus-type (hopping) charge carrier mechanism influenced the overall transport with an increase of the iodine concentration. This is represented as by the scheme as presented below:(5)$$I_{3}^{-}+I^{-}\to \;I^{-}\ldots \ldots I_{2}\ldots I^{-}\to I^{-}+I_{3}^{-}$$

Equation (5) demonstrates that when $I_{2}$ is switched from $I_{3}^{-}$ to $I^{-}$, $I^{-}$ and $I_{3}^{-}$ should be in a close immediacy to one other. Because both reactants are negatively charged, collisions between $I^{-}$ and $I_{3}^{-}$ are often problematic. Ionic liquids are made entirely of ions and have a relatively high molar concentration. It was discovered that triiodide could be transported to the counter-electrode not only by diffusion, but also by a non-diffusional hopping mechanism similar to that of Grotthus. Similar results were shown by Kawano et al. [54] who observed an increase in apparent diffusion coefficient (D_app_) of $I^{-}\mathrm{and}\;I_{3}^{-}$ with an increase in I_2_ concentration in ionic liquid compared to the normal solvent of the same viscosity. Figure 5 indicates that D_app_ depends on the concentration of the redox couple, and the value is larger for EMImTFSI than for polyethylene glycol diglycidyl ether (PEGDE).

## 9. Higher Ionic Conductivity

Enhancing the ionic conductivities of these GPEs is important and crucial for high DSSC conversion efficiency. Due to the high crystallinity of the polymers, traditional (solid) polymer electrolytes have relatively low ambient ionic conductivity. In this regard, the majority of recent research has focused on the synthesis and characterization of GPEs with increased ionic conductivity at room temperature. RTILs (room temperature ionic liquids) are ion sources as well as plasticizers. Highly conductive polymer gels made of a polymer matrix, plasticizer, and redox couple salts have been extensively explored to improve ionic conductivity to a practical level (at least 1 mS/cm). By mixing 5 wt% poly(vinylidenefluoride-co-hexafluoropropylene)(PVDF-HFP) with methoxypropionitrile (MPN)-based gel electrolytes, Wang et al. [37] created a series of quasi-solid-state DSSCs. At 1 Sun illumination, the conductivities of these polymer gels approached 10 mS/cm, and the cell efficiencies were over 6%. Cheng et al. [55] created a PVDF-based polymer gel system with a cross-linking reinforced network of polyethylene glycol dimethacrylate (PEGDMA), which had good ionic conductivity and mechanical toughness.

## 10. Excellent Thermal Stability

GPEs have exceptional thermal stability, and the DSSCs built on them have exceptional heat treatment stability. Ionic liquid-based electrolytes of poly(1-oligo (ethylene glycol) metha-crylate-3-methyl-imidazolium chloride) (P(MOEMImCl) containing 1-hexyl-3-methylidazolium iodide (HMImI) or a binary mixture of HMImI and 1-ethyl-3-methyl-imidazoliumtetrafluoroborate) (EMImBF4) showed minor weight loss at temperatures < 200 °C. The effect of heat treatments on DSSC performance based on this GPE showed drops in conversion efficiency of about 2.1 percent and 3.9 percent after heat treatments at 100 °C for 30 and 120 min, respectively, compared to the optimal efficiency of 6.1 percent at 30 °C for 5 min. This degradation during heat treatment was caused by iodine evaporation at elevated temperatures, which was confirmed by detecting evaporated iodine in an analyzer using wet starch paper during the heat treatment at 100 °C [56,57]. Dye desorption can also lead to a loss in cell performance at high temperatures according to the findings. Adsorption and desorption of dyes on TiO_2_ surfaces are in equilibrium, and this adsorption/desorption equilibrium alters with temperature [58]. In another work, an amphiphilic ruthenium sensitizer, cis-RuLL’(SCN)_2_, with a gel polymer electrolyte used to enhance the performance of a DSSC under thermal stress and light-soaking. It was observed that it produced an efficiency greater than 6% under full sunlight. In this scenario, heteroplastic ruthenium plays a key role in high temperature stability. In addition, it was observed that the cell maintained 94 percent of its initial functionality after 1000 h of heating at 80 °C. In a solar simulator (100 mW/cm^2^) equipped with a UV filter, the gadget also exhibited good stability after 1000 h of light soaking at 55 °C [59].

## 11. Outstanding Long-Term Stability

DSSCs containing the GPE have better long-term stability than DSSCs using liquid electrolytes. This is owing to the rapid devolatilization of liquid electrolytes and electrolyte leaks observed during their long-term operation. The results from prior work [60], as shown in Figure 6, shows that the efficiency of DSSC with the GPE PMMA–EC/PC/DMC–NaI/I_2_ declines by 8% after 5 days, while the efficiency of DSSC with liquid electrolyte drops by nearly 40%. After 40 days, the DSSC with a polymer gel electrolyte retains 83 percent of its initial light-to-electrical energy conversion efficiency, compared to only 27 percent for the DSSCs with liquid electrolytes [60].

## 12. Factors Influencing the Ionic Conductivity of GPE and the Photovoltaic Performance of Their DSSCs

The charge carrier transfer and diffusion efficiency of the redox couple resulting from its own ingredients, such as various types of polymers, the concentration of polymers with various molecular weights and conductivities, and the concentration and property of iodide salts are factors that influence the ionic conductivity of GPEs. Conductivity is influenced by external elements, such as organic solvents and temperature, to some extent. All of the aforementioned parameters influence the photovoltaic performance of DSSCs based on GPEs [61].

## 13. Approaches for the Enhancement in the Properties of GPE and Their DSSCs

Although utilizing GPEs increased the stability of DSSCs, the photovoltaic performance of GPE-based quasi-solid-state DSSCs was found to be lower than that of liquid-electrolyte-based DSSCs. Although some polymers may successfully gelate liquid electrolytes, they have a negative impact on photovoltaic performance and DSSC stability. For example, the gel network may obstruct charge transport in the gel electrolyte to some extent. Furthermore, the gelators may react with electrolyte components. To increase the ionic conductivity of GPEs and hence the performance of GPE-based DSSCs, the following procedures have been used.

(a)Thixotropic gel state: An appropriate mechanical tension can be used to convert thixotropic gels into sols. These thixotropic gel electrolytes should be useful for building DSSCs without leakage and giving efficient photovoltaic output that is stable over time [62].(b)Incorporation of proton donors: Due to their high proton conductivity, chemical and electrochemical stability, and ease of processing of polymer matrices, polymer electrolytes doped with proton donors have recently gained a significant amount of attention. The proton donor effectively increases the ionic conductivity of GPEs, resulting in increased $I^{-}/I_{3}^{-}$ mobility, short-circuit current density, open-circuit voltage, stability, and energy conversion efficiency of DSSCs [63,64].(c)Introduction of inorganic nanoparticles: Because the 3D network of the normal gel electrolyte hinders charge transport to some extent, adding inorganic nanoparticles could reduce charge combination at the interface of the dyed TiO_2_ electrode/electrolyte and increase the diffusion coefficient of $I_{3}^{-}$ because the introduction of inorganic nanoparticles reduces this negative effect [49].(d)Addition of pyridine derivatives: To improve the open-circuit photovoltage and therefore efficiency of DSSCs, additives in the electrolytes, such as pyridine derivatives, especially N-methylbenzimidazole, and tetrabutylammonium phosphate, are always added to the electrolytes [65,66].

## 14. Recent Developments in GPE-DSSC

The safety issues faced by the liquid electrolytes can be solved through the enhanced GPEs. Nonetheless, its efficiency of power conversion (η) can be improved through the betterment of the following parameters, viz. open-circuit voltage (V_OC_), short-circuit current (J_SC_) and fill factor (FF). Some research activities aim to improve GPEs. Saidi et al. [67] used a different concentration of 4-tert-butyl-pyridine (TBP) in a gel polymer (GP) for the performance improvement of DSSC and found that the quasi-fermi level of the TiO_2_ photo-anode shifted towards higher potential. The presence of 7% of TBP by weight increased V_OC_ by 21.31%. It is also noted that the addition of TBP showed a reduction in the J_SC_ of DSSC, while 3% of the TBP by weight showed the highest η of 8.11%. In another work, Praveen et al. [68] used chitosan dissolved in formic acid as an electrolyte for ZnO/ZnS-based DSSC, and reported a higher V_OC_ and η. This enhancement was attributed to higher ionic mobility, where ZnS overwhelms the charge combination rate, and chitosan helped in the phototronic effect as well as activating charge carrier for enhancing the ionic mobility and visible light absorption.

The PAN-co-PBA copolymer used as a gel electrolyte offers the following attributes: a carboxylate group preset in a PBA chain behaved as a superabsorbent to organic liquid, which impacted the ionic properties and long-life stability of the electrolyte. These properties provided support to the charge transportation between the electrolyte and conduction layer [69]. In another work, Chai et al. [70] used polyurethane acrylate (PUA) with tetrabutylammonium iodide (TBAI) as a gel-polymer electrolyte and found that there was an increment in electrolyte conductivity due to the high mobility and the number of densities of the charge carriers.

Abisharani et al. [71] obtained a high performance with better stability for DSSCs through the incorporation of the additives of the N, S, and O groups with GPE. The corresponding calculations of the DFT revealed that the anchoring groups of the additives play an important role in the charge transfer mechanism, and the TiO_2_ surface contained strong covalent/non-covalent bonds. In another work, Lobregas et al. [72] used potato starch modified with grafting 1-glycidyl-3-methylimidazolium chloride (GMIC) as a gel-polymer electrolyte, and obtained 0.514% efficiency and relative stability due to the good filling contact between the electrodes. Figure 7 shows the photocurrent–voltage curve characteristics of liquid and a modified gel–polymer electrolyte. Table 1 lists the performance characteristics of different modified gel polymers in DSSCs.

## 15. Conclusions

This article presented a review of the research on the development of electrolytes in general and the use of ionic liquids in particular for DSSCs. Though liquid electrolytes provide higher photovoltaic performance, the leakage and volatilization of the solvent reduced the enduring stability, making them impractical for large installations. Due to their stability, polymer electrolytes (solid and GPEs) and p-type semiconductors were chosen for study. The solid polymer electrolytes had low ionic conductivity and the p-type semiconductor had less efficiency and showed poor electrode contact. These made gel polymer electrolytes a suitable replacement over liquid electrolytes because of their higher ionic conductivity, enduring stability, and outstanding thermal stability of the DSSCs established on them. Furthermore, the ionic conductivity of GPE and the DSSCs performance can be improved through the concentrations of the iodide salts. However, in comparison to liquid electrolyte and GPE DSSCs, GPE persist lower ionic conductivity, photovoltaic performance (V_oc_), and energy conversion efficiency. Furthermore, the properties related to GPEs, viz. photovoltaic and ionic conductivity, can be improved by using strategies such as the thixotropic gel states, fusion of the proton donors, the addition of the pyridine derivatives GPEs, and the introduction of inorganic nanoparticle.

These strategies, combined with the unique properties of ionic liquids incorporated into GPEs, have the potential to expand the scope of GPE to obtain higher efficiency, high ionic conductivity, and continuous industrially manufactured DSSCs.

## Figures and Tables

**Figure 1 micromachines-13-00680-f001:**
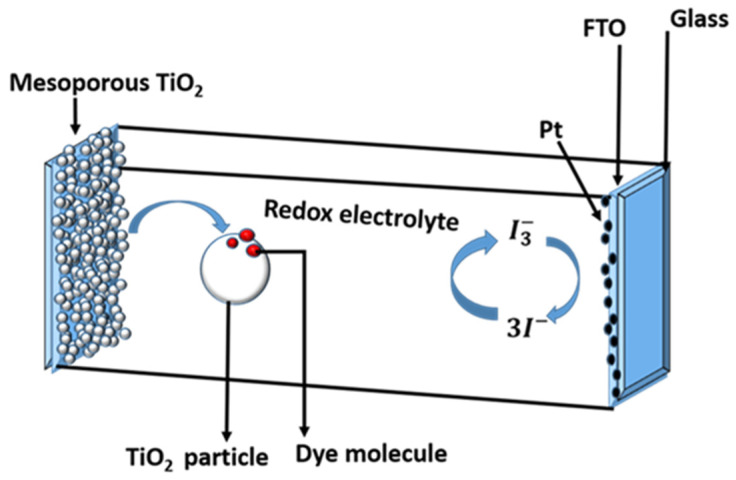
Schematic of the interior components of a dye-sensitized solar cell on fluorine-doped tin oxide (FTO) coated glass substrate.

**Figure 2 micromachines-13-00680-f002:**
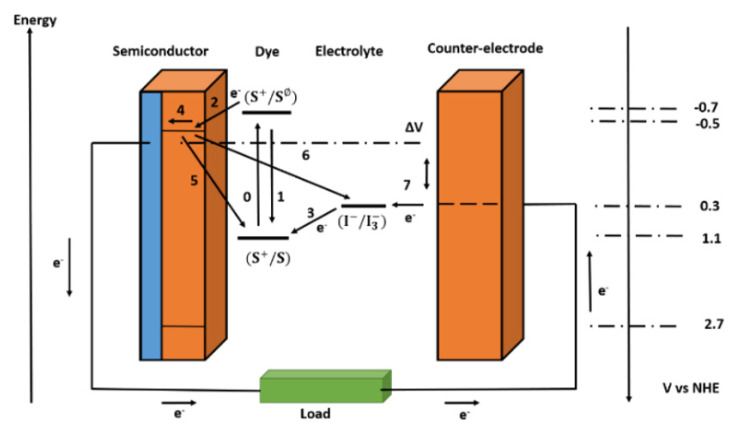
Simple energy level diagram for a DSSC. The basic electron transfer processes are indicated by numbers (1–7).

**Figure 3 micromachines-13-00680-f003:**
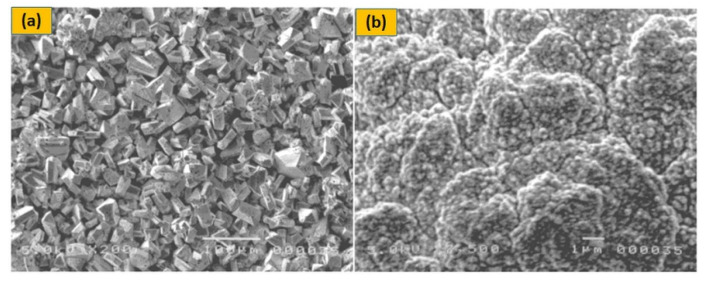
SEM images of CuI crystals deposited on the dyed TiO_2_ porous film: (**a**) CuI without molten salt; (**b**) CuI/1-methyl-3-ethyl-imidazolium thiocyanate composite electrolyte, adapted from reference [42]. [Reproduced with kind permission].

**Figure 4 micromachines-13-00680-f004:**
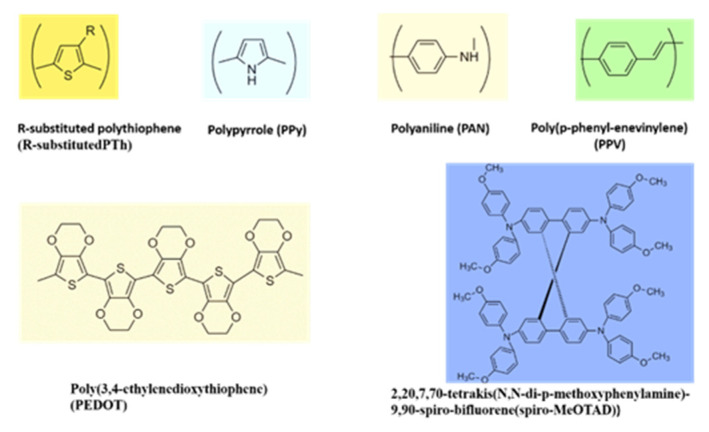
Chemical structures of a few conducting polymers with hole transport properties.

**Figure 5 micromachines-13-00680-f005:**
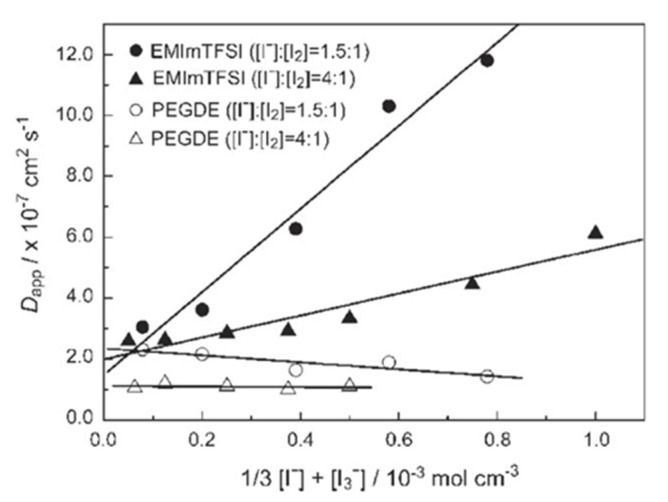
Relationship between D_app_ and the concentration of the redox couple with different molar ratios dissolved in EMImTFSI and PEGDE, adapted from reference [54].

**Figure 6 micromachines-13-00680-f006:**
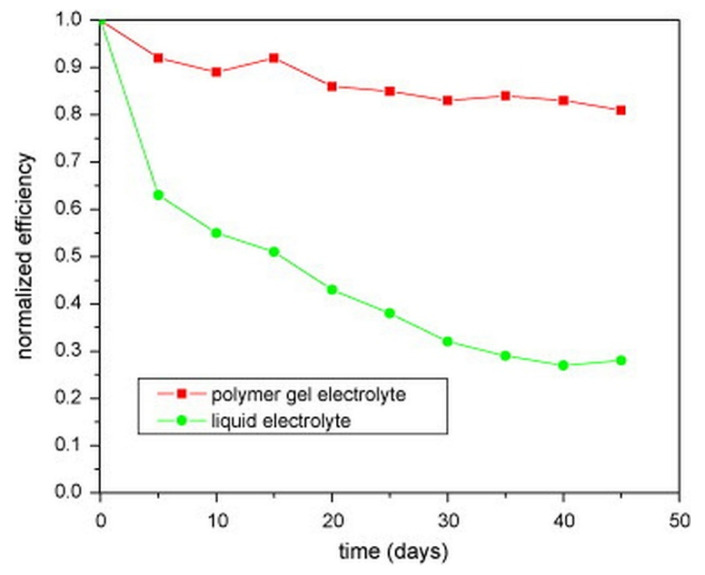
Long-term stability of the DSSCs with liquid electrolyte (circle) and with the gel polymer electrolyte PMMA–EC/PC/DMC–NaI/I_2_ (square), adapted from reference [60]. [Reproduced with kind permission].

**Figure 7 micromachines-13-00680-f007:**
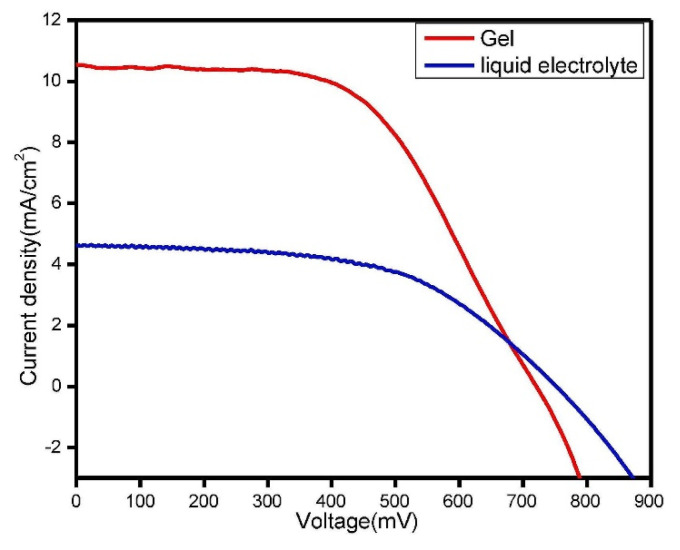
Under the illumination of simulated solar light, photocurrent voltage curves for devices containing liquid electrolyte and GG (0.6 g)-modified GPEs were measured, and adapted from reference [73]. [Reproduced with kind permission].

**Table 1 micromachines-13-00680-t001:** List of performance characteristics of various modified gel-polymer DSSCs.

S. No	Author	Electrolyte	V_OC_ (V)	J_SC_ (mA cm^−2^)	FF (%)	η (%)
1	Praveen et al. [68]	Chitosan	0.8	1.6871	44.59	1.6
2	Shah et al. [69]	7% PAN-co-PBA	0.646 ± 0.03	13.16 ± 0.71	61.5 ± 2	5.23 ± 0.28
3	Chai et al. [70]	67.94%PUA–30.00% TBAI–2.06% I_2_	0.55 ± 0.01	7.15 ± 0.74		1.97 ± 0.21
4	Rao et al. [74]	PEO/PEGDME/0%acetamide	0.79	11.35	0.55	5.03
5	Abisharani et al. [71]	SAA/I^−^/I_3_^−^/Gelatin	0.79	14.1	0.52	5.8
6	Gunasekaran et al. [73]	0.6-Guar gum	0.787	10.65	0.46	4.96
7	Manafi et al. [75]	PVDF– HFP/PEO/BMIMBF_4_ (60/40 wt%)	0.685	15.65	60.4	6.47
8	Balamurgan et al. [76]	Co^2+^/^3+^[bnbip]_2_/HEC/BNBIT	0.795	10.7	0.53	4.50
9	Khannam et al. [77]	Gelatin/Graphene Oxide/LiI/TBP/MPI/NMP	0.75	7.68	0.7	4.02
10	Sharma et al. [78]	Gelatin/MWCNT/LiI/I2/TBP/MPI/NMP	0.93	8.14	0.18	1.35
11	Farhana et al. [79]	P(VB-co-VA-co-VAc)/NaI	0.61	12.52	51.8	4.01
12	Farhana et al. [80]	P(VB-co-VA-co-VAc)/TPAI	0.678	13.585	50.1	4.615
13	Careem et al. [81]	50%Ki-50%TPAI-PVA	0.630	8.0	62	5.51
14	Zulkifi et al. [82]	PhCh:EC:DMF:KI/I2	0.37	20.33	65	3.57
15	Kesavan et al. [83]	Au_97.5_ Pt_2.5_/C NPoS	0.686	13.09	56.8	5.1
16	Suzuka et al. [84]	Indolines (2,3-benzo-4,5-dihydroindoles)	0.93	15.5	70	10.1
17	Lin et al. [33]	POEI-IS	0.825	13.85	71	8.18
18	Xiang et al. [31]	[Co(bpy_3_)] ^3+^	0.817	1.54 ± 0.01	80	10

## Data Availability

Not applicable.

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
