# Peer review of "A Review on Gel Polymer Electrolytes for Dye-Sensitized Solar Cells"

_micromachines, 2022, doi:10.3390/mi13050680_

Round 1
Reviewer 1 Report
This review summarizes polymer gels in using DSSCs.
In the practical application of DSSC, the use of liquid electrolyte leads to a decrease in durability. And therefore, related research has been conducted to replace the electrolyte with ionic liquids or polymers. This review summarizes those studies and is considered to be a very useful review.
The energy levels of DSSC for liquid electrolytes are described.
However, for solid electrolytes and p-type semiconductors, the principles of energy levels and the mechanism of DSSC using solid electrolyte or p-type semiconductors are not described in this paper. In fact, those principles are considered to be extremely important and they need to be explained in this review.
Reviewer 2 Report
A Review on Gel Polymer Electrolytes for Dye-Sensitized Solar Cells
Author worked on a review based gel polymer electrodes for dye sensitized solar cells. This review is written in a fluent and good way for readers. However, I suggest literature citation for some parts.
- In introduction part: I suggest two papers citation,
*Osbel Almora et al., Device Performance of Emerging Photovoltaic Materials (Version 2), Advanced Energy Materials, https://doi.org/10.1002/aenm.202102526, 2021.
*Osbel Almora et al., Device Performance of Emerging Photovoltaic Materials (Version 1), Adv. Energy Mater. 2020, 2002774 https://onlinelibrary.wiley.com/doi/pdf/10.1002/aenm.202002774
- There is not enough references in Part 2 “Dye-Sensitized Solar Cell Architecture and Basic Operation”. I suggest following paper to be cited.
* Yuji Kubo et al., Synthesis of a dibenzo-BODIPY-incorporating phenothiazine as a panchromatic sensitizer for dye-sensitized solar cells, New J. Chem., 2017, 41, 10367-10375, DOI: 10.1039/C7NJ01735A, Paper
http://pubs.rsc.org/en/content/articlelanding/2017/nj/c7nj01735a#!divAbstract
* Michael Graetzel et al. Optimization of Distyryl-Bodipy Chromophores for Efficient Panchromatic Sensitization in Dye Sensitized Solar Cells
Chemical Science; Vol.2, Issue. 5, Pages 949-954, 2011; DOI:10.1039/C0SC00649A
- In part 5. “P-type semiconductors”,
There are some p-type semiconductors that published their application for the first time that I suggest citation to these papers.
*Yuta Higashino et al., π-Expanded dibenzo-BODIPY with near-infrared light absorption: Investigation of photosensitizing properties of NiO-based p-type dye-sensitized solar cells Dyes and Pigments, Volume 170, November 2019, Article 107613, https://www.sciencedirect.com/science/article/abs/pii/S0143720819310526
* Fırat Sahiner et al., Naphthalene imides as novel p-type sensitizers for NiO-based dye-sensitized solar cells, https://doi.org/10.1039/D0NJ03266B , New Journal of Chemistry , 2020 , https://pubs.rsc.org/en/content/articlelanding/2020/nj/d0nj03266b#!divAbstract
- In part 10 “ Excellent thermal stability”,
I suggest some ionic liquid papers’ citation
* Serpil Denizalti et al., Dye-Sensitized Solar Cells using Ionic Liquids as Redox Mediator, Chemical Physics Letters 691 (2018) 373–378, https://doi.org/10.1016/j.cplett.2017.11.035
*Synthesis of phenanthrene-annulated imidazolium electrolyte-based dye sensitized solar cells as redox mediators, https://doi.org/10.1016/j.tsf.2020.138372 , Thin Solid Films ,2020, https://www.sciencedirect.com/science/article/abs/pii/S0040609020305812
Reviewer 3 Report
In this review paper, the advantages of gel polymer electrolytes (GPEs) are discussed along with other type of electrolytes such as solid polymer electrolytes and p-type semiconductor-based electrolytes. It is well written and could be accepted for publication in Micromachines. It only needs to check English and the format of references.
Reviewer 4 Report
The manuscript reviews progress on polymer gel electrolytes for dye-sensitized solar cells (DSSCs). The article is well organized, clearly written and appropriately illustrated. It gives a solid background on the field in the first parts and a good overview of recent developments on DSSC electrolytes in the later parts.
What I think that is insufficiently addressed in the present manuscript is the work on other types of electrolytes, to compare and contrast with the more recent reports. I have in mind in particular the following papers:
- Peng Wang et al., Nature Materials 2003, 2:402 (doi:10.1038/nmat904) on a stable quasi-solid-state PV cell with a polymer gel electrolyte;
- SM Zakeeruddin and M Grätzel, Adv. Funct. Mater. 2009, 19:2187 (DOI: 10.1002/adfm.200900390), a review on solvent-free ionic liquid electrolytes;
- Jun-Ho Yum et al., Nature Communications 2012, 3:631 (DOI: 10.1038/ncomms1655), on a cobalt complex redox shuttle for DSSCs with high open-circuit potentials;
- Mingkui Wang et al., Energy Environ. Sci. 2012, 5:9394 (DOI: 10.1039/c2ee23081j), a review on redox electrolytes for DSSCs with a focus on increasing photovoltaic conversion efficiency by fine tuning the open-circuit voltage.
I strongly believe that these references, not cited in the manuscript, are important contributions in the field that ought to be mentioned in the review. The idea of stability, which is developed in the present manuscript reflects a very important aspect in the field of DSSCs. However, to offer a more thorough description of the topic, it needs to be complemented with at least two other important criteria. One is the proper alignment of the redox level with respect to the ground state of the dye (to allow for sensitizer regeneration) and the other the distance from the Fermi level of the photoanode, (which affects the open-circuit voltage and the overall PV conversion efficiency).
Reviewer 5 Report
In general, this review is well written and can be considered for publication after a minor revision. However, I cannot understand why this is submitted to Micromachines. This work is more suitable for Energy related journal.
Detailed comments to authors:
- Fig. 1 does not show any data which is not presented in Fig. 2. I propose to remove Fig. 1 from manuscript or combine it with Fig. 2.
- If any image shown in this work is adopted from other publication it should be clearly mentioned. For example: “adapted from ref. xxx with the permission of publisher (name of publisher)”
- More information on iodide-free redox system should be provided. Page 5 line 210. Authors mentioned “Alternative redox systems, such as cobalt-based systems, SCN- /(SCN) 3- and SeCN- / (SeCN) 3-, have shown promising results in recent investigations”. However, examples of those works are not presented in this review. Those results are promising, as it was mentioned by authors, and should be highlighted here. The performance data on most promising system should be added to Table 2. See example of papers related with Co-based redox:
Chem. Commun. 2015, 51, 16308
Chem. Commun. 2013, 49, 8997
ChemSusChem 2015, 8, 3704
Example of alternative (to iodine-based) redox systems:
ACS Appl. Mater. Interfaces. 2016;8:15267
Sci. Rep. 2016;6:28022
